# Variation and internal-external driving forces of grey water footprint efficiency in China's Yellow River Basin

**Yun Li[1], Yu Liu[1]\*, Lihua Yang[1], Tianbo Fu[2]**

**1** Business school, Hohai University, Nanjing, China, **2** Business school, Jiangsu Open University, Zhenjiang, China

\* liuyuhhu@outlook.com

**Data Availability Statement:** All relevant data and the sources have been provided in the paper and the Supporting Information file.

**Funding:** This study is supported by the "Graduate Research and Innovation Projects of Jiangsu

## Abstract

Grey water footprint (GWF) efficiency is a reflection of both water pollution and the economy. The assessment of GWF and its efficiency is conducive to improving water environment quality and achieving sustainable development. This study introduces a comprehensive approach to assessing and analyzing the GWF efficiency. Based on the measurement of the GWF efficiency, the kernel density estimation and the Dagum Gini coefficient method are introduced to investigate the spatial and temporal variation of the GWF efficiency. The Geodetector method is also innovatively used to investigate the internal and external driving forces of GWF efficiency, not only revealing the effects of individual factors, but also probing the interaction between different drivers. For demonstrating this assessment approach, nine provinces in China's Yellow River Basin from 2005 to 2020 are chosen for the study. The results show that: (1) the GWF efficiency of the basin increases from 23.92 yuan/m$^3$ in 2005 to 164.87 yuan/m$^3$ in 2020, showing a distribution pattern of "low in the western and high in the eastern". Agricultural GWF is the main contributor to the GWF. (2) The temporal variation of the GWF efficiency shows a rising trend, and the kernel density curve has noticeable left trailing and polarization characteristics. The spatial variation of the GWF efficiency fluctuates upwards, accompanied by a rise in the overall Gini coefficient from 0.25 to 0.28. Inter-regional variation of the GWF efficiency is the primary source of spatial variation, with an average contribution of 73.39%. (3) For internal driving forces, economic development is the main driver of the GWF efficiency, and the interaction of any two internal factors enhances the explanatory power. For external driving forces, capital stock reflects the greatest impact. The interaction combinations with the highest q statistics for upstream, midstream and downstream are capital stock and population density, technological innovation and population density, and industrial structure and population density, respectively.

## Introduction

Water serves as a crucial material basis that sustains the human society. During rapid economic and social development, problems such as the over-consumption of water resources

Province" (grant number: KYCX21_0441) and the "Fundamental Research Funds for the Central Universities" (grant number: B220203024). The funder has played an important role in study design, data collection and analysis, decision to publish, and preparation of the manuscript.

**Competing interests:** The authors have declared that no competing interests exist.

and damage to the water environment have occurred frequently, resulting in the water crisis around the world [1,2]. A quantitative assessment of water pollution is an important prerequisite for alleviating the water crisis [3]. The grey water footprint (GWF) theory, first proposed by Hoekstra and Chapagain [4], provides a new approach to measuring the quality of polluted water. GWF is defined as the volume of freshwater required to absorb a given pollutant [5–7]. However, sustainable development needs to accommodate both economic and environmental co-benefits [8,9]. The simple concept of the GWF expresses the impact of pollution on the volume of water resources, but does not capture the economic role. As a result, some scholars have further incorporated the role of the economy into the measurement of water pollution by combining the GWF with GDP, which constitutes the GWF efficiency [10–13]. A comprehensive analysis of the GWF and its efficiency is an essential basis for reconciling the relationship between the environment and the economy.

As the second longest river in Asia and the " mother river " of China, the Yellow River supplies water to Asia's major agricultural production areas, such as the Hetao Plain and the Fenwei Plain. Moreover, the Yellow River Basin is an important economic belt for the energy industry [14]. However, over the years, the water quality of the basin has been heavily polluted. With the rise of industry and the accelerated urbanization of the population, wastewater from industry and urban dwellers has increased dramatically [15]. In addition, the over-application of pesticides and fertilizers in the growing areas of the basin is also a severe problem [16]. Consequently, the Yellow River carries about 6% of wastewater and 7% of chemical oxygen demand with 2% of China's water resources, significantly exceeding the carrying capacity of the water environment [17]. At the same time, problems such as unbalanced economic development also pose a huge challenge to the sustainable development of the Yellow River Basin [18]. Therefore, this study takes the Yellow River Basin as an example to measure the GWF and its efficiency, and to accurately grasp the variation and driving forces of water pollution and economic development in the basin at the spatial and temporal levels, which is representative and relevant.

A series of studies have been conducted on the GWF of the agricultural [7,19–21], industrial [22,23] or domestic sectors [24,25]. Many scholars have analyzed the GWF or its efficiency at the national [3,10,26,27], regional [28–31] or river basin level [11–13,32,33] from the perspective of quantitative measurements. According to the results, it is not difficult to find that the GWF and its efficiency show some numerical variability at the spatial and temporal levels. However, these simple comparisons of visual data often ignore the specific spatial and temporal distribution of subsamples. Few studies have been conducted to quantitatively and methodologically investigate the spatial and temporal variation of the GWF and its efficiency. In this study, the kernel density estimation and the Dagum Gini coefficient method are used to systematically and comprehensively investigate the spatial and temporal variation of the GWF efficiency in the Yellow River Basin. In terms of temporal variation, the kernel density estimation method is able to reflect the continuous distribution characteristics of the data by obtaining estimates of each point of the density function [34,35]. In terms of spatial variation, existing researches tend to use the traditional Gini coefficient method or the Thiel index method to analyze the evolution of the GWF, which cannot solve the overlap problem between sub-samples [36,37]. In contrast, the Dagum Gini coefficient method can effectively overcome this problem and decompose the spatial differences, thus accurately reflecting the spatial variation and its sources [18].

In addition, exploring the driving forces of spatial and temporal changes in the GWF efficiency is a priority for reconciling economic and social development with ecological conservation. Currently, there is a wealth of research related to the driving force of GWF or its efficiency. For example, Zhang and Sun [38], Han et al. [10], Zhang et al. [3], and Feng et al.

[27] all use the Kaya equation and LMDI model to dissect the effects of multiple factors on the GWF or its efficiency of different provinces in China. On this basis, Fu et al. [12], Chen et al. [11], and Xu et al. [13] decompose the GWF efficiency into several driving factors by the same method with different watersheds as the object of their study. Bai and Sun combine the Thiel index and the extended Kaya equation to explore the regional differences and driving factors of the GWF per capita [37]. However, the Kaya equation and the LMDI decomposition method used in most studies can quantify the contribution of driving factors to the GWF efficiency, but cannot investigate the interaction of different driving factors. In contrast, the Geodetector method is able to identify the driving factors and their interaction relationships behind the spatial and temporal variation [18,39]. Therefore, this method is selected in this study to analyze the internal-external driving forces of spatial and temporal variation in the GWF efficiency in the Yellow River Basin.

This study aims to introduce an integrated assessment approach for the GWF efficiency. Based on the measurement of the GWF and its efficiency in the 9 provinces of the Yellow River Basin from 2005 to 2020, the temporal and spatial variation of GWF efficiency is investigated by the kernel density estimation and the Dagum Gini coefficient method, respectively. The internal and external driving forces and their interaction of the GWF efficiency are also discussed with the help of the Geodetector. Finally, targeted suggestions are put forward for improving the GWF efficiency. The contributions of this study are: on the one hand, a comprehensive assessment of GWF efficiency is provided at the watershed scale. It also takes into account both temporal and spatial dimensions to reveal the variation in the evolution of GWF efficiency, which can be used to clearly observe the pollution and economic development of the Yellow River Basin. This not only expands the research thinking on GWF efficiency but also contributes to the integrated management of the river basin. On the other hand, the Geodetector method has been innovatively applied to investigate the individual and interactive driving forces of the internal and external factors on GWF efficiency, with a view to finding the main driving factors for the variation of GWF efficiency in different regions. It is useful for the zoning control of water pollution, optimal regulation of water resources and sustainable ecological development.

## Overview of the study area

The Yellow River Basin in China spans nine provinces, connecting the Qinghai-Tibet Plateau, the Loess Plateau and the North China Plain, making it an important ecological and economic zone. With only 27% of China's average per capita water resources, the Yellow River Basin is responsible for supplying water to 12% of the country's population, 17% of its arable land and more than 50 large and medium-sized cities. In addition, the Yellow River Basin is an important industrial production area in China, with abundant traditional resources such as coal, oil, natural gas and non-ferrous metals. However, the rough development has led to excessive consumption of resources and severe water pollution problems in the river basin. In 2022, the report of the 20th National Congress of the Communist Party of China (CPC) put forward "promoting the ecological protection and high-quality development of the Yellow River basin" as a major strategy to facilitate coordinated regional development. Although the Yellow River has been managed to a certain extent, the problem of fragile water ecology in the basin is still prominent. Coordinating the relationship between water pollution and economic development is a must for the implementation of high-quality development in the Yellow River Basin.

Therefore, nine provinces in China's Yellow River Basin, including Qinghai, Sichuan, Gansu, Ningxia, Inner Mongolia, Shanxi, Shaanxi, Henan and Shandong, are taken as the research objects. The GWF and its efficiency are measured for the nine provinces between 2005 and 2020. Fig 1 shows an overview of the study area.

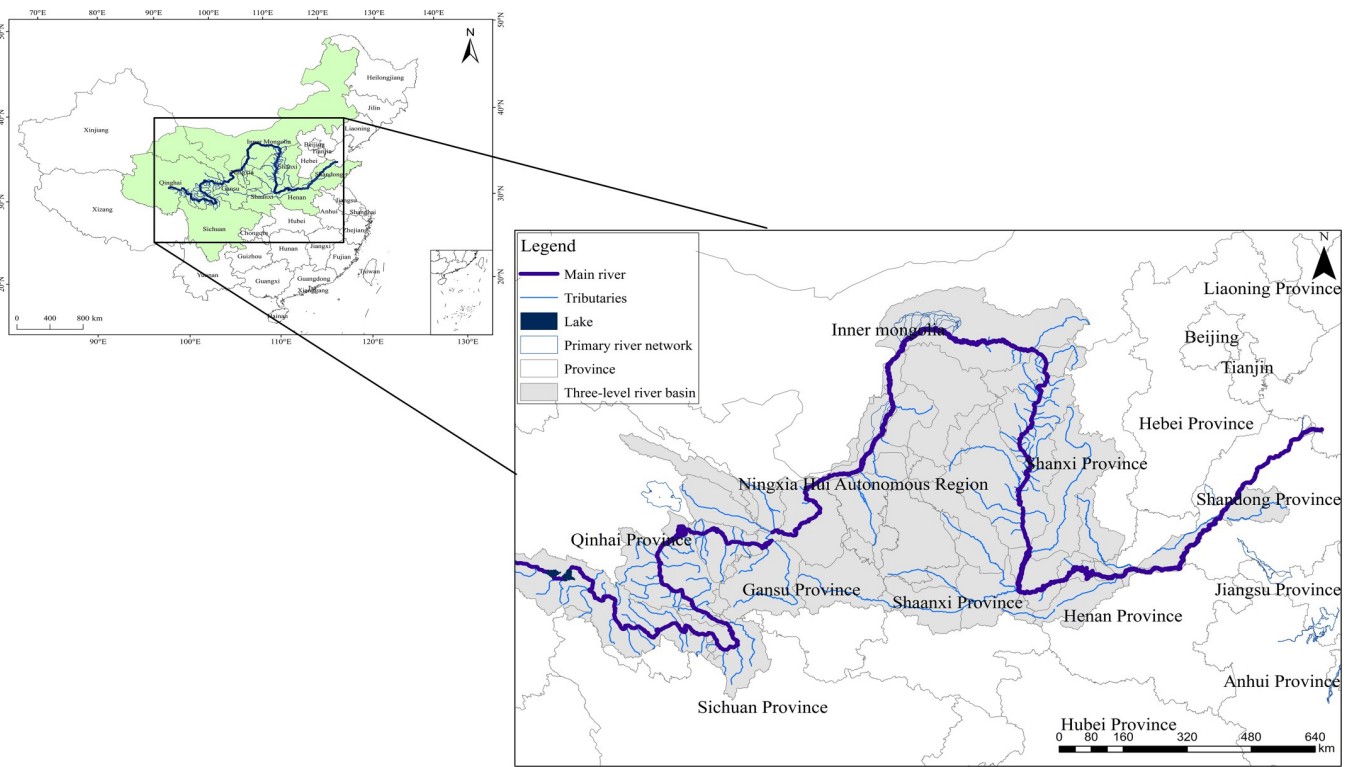

**Fig 1. Overview of the study area.**

## Methods and data

### GWF efficiency accounting

This study measures the GWF efficiency of the Yellow River Basin in China following the evaluation methodology presented in the *Water Footprint Assessment Manual* by Hoekstra et al. [40]. The accounting is based on three aspects: agricultural, industrial and domestic GWF.

**Agricultural GWF.** The GWF from planting and breeding are measured separately to obtain the agricultural GWF. The improper use of pesticides and fertilizers in the planting process has caused serious pollution of water. In this study, the degree of water pollution by nitrogen fertilizers is chosen as the evaluation indicator for the GWF from planting. The specific formula is as follows.

$$GWF_{pla} = \frac{\alpha * Appl}{C_{\max} - C_{nat}} \tag{1}$$

Where $GWF_{pla}$ is the GWF from planting; $\alpha$ represents the proportion of nitrogen fertilizer entering the water; $Appl$ is the annual use of nitrogen fertilizer; $C_{\max}$ and $C_{nat}$ indicate the concentration of the pollutant required in the standard and the background concentration of the pollutant, respectively.

The pollutants chemical oxygen demand (*COD*) and total nitrogen (*TN*) are both used as the basis for the evaluation of the GWF from breeding, calculated as follows.

$$GWF_{bre} = \max(GWF_{bre(COD)}, GWF_{bre(TN)}) \tag{2}$$

$$GWF_{bre(i)} = \frac{L_{bre(i)}}{C_{\max} - C_{nat}} \qquad (3)$$

$$L_{bre(i)} = \sum_{h=1}^{4} N_h * D_h * (f_h * p_{hf} * \beta_{hf} + u_h * p_{hu} * \beta_{hu}) \qquad (4)$$

Where $GWF_{bre}$ is the GWF from breeding; $i$ is the pollutant $COD$ or $TN$; $GWF_{bre(i)}$ represents the GWF from the breeding of different pollutants; $L_{bre(i)}$ represents the pollutant load of breeding pollutants; $h$ is different livestock (pigs, cattle, sheep, poultry); $N_h$ and $D_h$ are the number and the breeding cycle of $h$; $f_h$ and $u_h$ are the daily manure discharge and the daily urine discharge of $h$; $p_{hf}$ and $p_{hu}$ are the pollutant content per unit of manure and urine for $h$; $\beta_{hf}$ and $\beta_{hu}$ represent the pollutant loss rate per unit of manure and urine for $h$, respectively.

According to the metric model for the GWF from planting and breeding, the agricultural GWF ($GWF_{agr}$) can be calculated by the following equation:

$$GWF_{agr} = \max(GWF_{bre(COD)}, GWF_{pla} + GWF_{bre(TN)}) \qquad (5)$$

**Industrial GWF.** Chemical oxygen demand ($COD$) and ammonia nitrogen ($NH_4^+ - N$) are the main pollutants in the industrial effluent, which are selected as indicators to measure the industrial GWF. Then

$$GWF_{ind} = \max(GWF_{ind(COD)}, GWF_{ind(NH_4^+ - N)}) \qquad (6)$$

$$GWF_{ind(i)} = \frac{L_{ind(i)}}{C_{\max} - C_{nat}} \qquad (7)$$

Where $GWF_{ind(i)}$ represents the industrial GWF of the pollutant $i$ ($COD$ or $NH_4^+ - N$). $L_{ind(i)}$ is the discharge load of the pollutant $i$.

**Domestic GWF.** Domestic wastewater is similar to industrial wastewater in that $COD$ and $NH_4^+ - N$ are the main pollutants, so the domestic GWF ($GWF_{dom(i)}$) is calculated in the same way as for industry.

**GWF efficiency.** The total GWF ($TGWF$) can be obtained by summing the agricultural, industrial and domestic GWF.

$$TGWF = GWF_{agr} + GWF_{ind} + GWF_{dom} \qquad (8)$$

On this basis, the GWF efficiency ($g$), i.e., the GDP that can be generated from one unit of GWF, can be further calculated as:

$$g = \frac{GDP}{TGWF} \qquad (9)$$

## Kernel density estimation

In this study, kernel density estimation is used to explore the dynamic evolution of GWF efficiency in the temporal dimension. As one of the non-parametric estimation methods, the core idea of kernel density estimation is to characterize the distribution pattern of random variables

through a continuous density profile [34,35]. The density function equation is as follows.

$$f(x) = \frac{1}{Nh} \sum_{i=1}^{N} K\left(\frac{X_i - \bar{X}}{h}\right) \tag{10}$$

Where $N$ is the number of sample observations; $X_i$ is the independent and identically distributed observations, i.e., the GWF efficiency of the 9 provinces in the Yellow River Basin; $\bar{X}$ is the mean of the GWF efficiency; $h$ represents the bandwidth, reflecting the degree of smoothness and estimation accuracy of the density function curve; $K$ represents the kernel density and the kernel function $K(\bullet)$ is expressed as follows.

$$K(x) = \frac{1}{\sqrt{2\pi}} \exp\left(-\frac{x^2}{2}\right) \tag{11}$$

$$\begin{cases} \lim_{x \to \infty} K(x) * x = 0 \\ K(x) \geq 0, \int_{-\infty}^{+\infty} K(x)dx = 1 \\ \sup K(x) < +\infty, \int_{-\infty}^{+\infty} K^2(x)dx = 1 \end{cases} \tag{12}$$

## Dagum Gini coefficient and its decomposition

The Dagum Gini coefficient reflects the changes in the relative variation of indicators. It is used in this study to measure the regional variation of the GWF efficiency in the Yellow River Basin. Drawing on Dagum and Alvaredo [41,42], the 9 provinces are divided into three sub-regions: upstream, midstream and downstream. Among them, upstream includes Qinghai, Sichuan, Gansu and Ningxia; midstream includes Inner Mongolia, Shanxi and Shaanxi; and downstream includes Henan and Shandong. $j$ and $h$ denote different sub-regions; $n_j$ and $n_h$ represent the number of provinces in sub-regions $j$ and $h$; $y_{ji}$ and $y_{hr}$ represent the GWF efficiency of the province $i$ in region $j$, and the GWF efficiency of province $r$ in region $h$, respectively; $\bar{y}$ is the mean of GWF efficiency. Then, the overall Gini coefficient can be expressed as:

$$G = \frac{\sum_{j=1}^{k} \sum_{h=1}^{k} \sum_{i=1}^{n_j} \sum_{r=1}^{n_h} |y_{ji} - y_{hr}|}{2n^2\bar{y}} \tag{13}$$

On this basis, the Dagum Gini coefficient is decomposed to investigate the sources of variation in the GWF efficiency of different sub-regions. According to the sub-sample decomposition method, the sources of overall variation can be decomposed into three components: the contribution of intra-regional variation ($G_w$), inter-regional variation ($G_{nb}$), and hypervariable density ($G_t$). $G = G_w + G_{nb} + G_t$. $G_{jj}$ and $G_{jh}$ represent the Gini coefficient within the sub-region $j$, and the Gini coefficient between sub-region $j$ and $h$, respectively. The specific formulas are as follows.

$$G_{jj} = \frac{\sum_{i=1}^{n_j} \sum_{r=1}^{n_h} |y_{ji} - y_{jr}|}{2n_j^2\bar{y}_j} \tag{14}$$

$$G_{jh} = \frac{\sum_{i=1}^{n_j} \sum_{r=1}^{n_h} |y_{ji} - y_{hr}|}{n_j n_h (\bar{y}_j + \bar{y}_h)} \tag{15}$$

$$G_w = \sum_{j=1}^{k} G_{jj} p_j s_j \tag{16}$$

$$G_{nb} = \sum_{j=2}^{k} \sum_{h=1}^{j-1} G_{jh}(p_h s_j + p_j s_h) D_{jh} \tag{17}$$

$$G_t = \sum_{j=2}^{k} \sum_{h=1}^{j-1} G_{jh}(p_h s_j + p_j s_h)(1 - D_{jh}) \tag{18}$$

Where $p_j = n_j/n$, $s_j = n_j \bar{y}_j / n\bar{y}$, $j = 1,2,\cdots,k$; $D_{jh}$ represents the relative effect of GWF efficiency between sub-region $j$ and $h$; $d_{jh}$ represents the difference in GWF efficiency between regions and $q_{jh}$ is the hypervariable first order moment. Then

$$d_{jh} = \int_0^\infty dF_j(y) \int_0^y (y - x) dF_h(x) \tag{19}$$

$$q_{jh} = \int_0^\infty dF_h(y) \int_0^y (y - x) dF_j(x) \tag{20}$$

Where $F_j$ and $F_h$ represent the cumulative density distribution functions for the region $j$ and $h$.

## Decomposition of driving factors

Geodetectors include factor detection, interaction detection, risk detection and ecological detection [43]. In this study, factor detection and interaction detection are used to explore the internal-external driving forces of the single factor alone and the interaction between different factors on the variation of GWF efficiency, respectively.

Factor detection. $q$ values reflect the effect on grey water footprint efficiency, $q \in [0,1]$. $q$ values indicate that $X_i$ explains $100^* q\%$ of $Y$, i.e., the higher the $q$ value, the greater the explanatory power of variable $X_i$ on the variation in GWF efficiency. $q$ is expressed as:

$$q = 1 - \frac{1}{N\sigma^2} \sum_{h=1}^{m} N_h * \sigma_h^2 \tag{21}$$

Where $h = 1,2,\cdots,m$, is the strata of the independent or dependent variable. $N$ and $N_h$ are the sample sizes for the whole basin and strata, respectively. $\sigma^2$ and $\sigma_h^2$ represent the variance of the dependent variable for the whole basin and strata, respectively.

Interaction detection. Interactions between different factors are identified by assessing whether factors $X_i$ and $X_j$ acting together increase or decrease the explanatory power ($q$ value) of the dependent variable $Y$. Specifically, if $q(X_i \cap X_j) < \min(q(X_i), q(X_j))$, the interaction of $X_i$ and $X_j$ is non-linearly weakened; if $\min(q(X_i), q(X_j)) < q(X_i \cap X_j) < \max(q(X_i), q(X_j))$, the interaction of $X_i$ and $X_j$ is one-way non-linearly weakened; if

$\max(q(X_i), q(X_j)) < q(X_i \cap X_j) < q(X_i) + q(X_j)$, the interaction of $X_i$ and $X_j$ is two-way strengthened; if $q(X_i \cap X_j) = q(X_i) + q(X_j)$, it means that $X_i$ and $X_j$ are independent of each other; if $q(X_i \cap X_j) > q(X_i) + q(X_j)$, the interaction of $X_i$ and $X_j$ is non-linearly strengthened.

### Data sources

**Variables of GWF efficiency.** This study measures the GWF efficiency of different sectors based on the statistical data from 2005–2020 in the 9 provinces of the Yellow River Basin. Among them, data on the annual use of nitrogen fertilizer in the agricultural sector are obtained from the *Agricultural Statistics of New China in the Past Fifty Years*, and the proportion of nitrogen fertilizer entering water bodies is 7%; data on the number of livestock, breeding cycle, daily manure and urine discharge, pollutant content per unit of manure and urine, and pollutant loss rate per unit of manure and urine are obtained from the *China Rural Statistical Yearbook* and the *China Pollution Technical Report of the Survey of Large-scale Livestock and Poultry Breeding Industry*; data of industrial and domestic pollutant discharges and GDP can be obtained from the *China Environmental Statistical Yearbook* and *China Statistical Yearbook*. In addition, with reference to Hoekstra et al. [40], Han et al. [10], Zhang et al. [3] and the *Integrated Pollutant Discharge Standard (GB8978-1996)*, the background concentration of the pollutant is set at 0, and the concentrations of COD and nitrogen required in the standard are 60 mg/L and 15 mg/L, respectively.

**Variables of the driving forces.** This study investigates the key driving forces of the spatial and temporal variation of GWF efficiency in the Yellow River Basin from both internal and external perspectives. On the one hand, the underlying cause of the variation in GWF efficiency is the correlation and mutual feedback between the internal forces that constitute the GWF efficiency. The economic development factor, represented by GDP, is one of the significant drivers of GWF efficiency. As the measurement of GWF efficiency is dependent on water use and pollution emissions, the factors of water consumption and pollution levels should also be included as internal driving forces. In addition, labour and environmental capital inputs can influence grey water output in different industries which are also critical internal drivers that cannot be ignored. Therefore, in this study, economic development (*ED*), water consumption (*WC*), pollution level (*PL*), labour input (*LI*) and environmental capital input (*EI*) are selected as internal driving forces affecting GWF efficiency. The economic development is measured by the GDP of each province; the water consumption is represented by the ratio of water consumption to total water resources; the pollution level is based on the ratio of total COD and ammonia nitrogen emissions to water consumption in different sectors; the labour input is measured by the employment rate of the urban population; and the environmental capital investment is measured by the amount of investment in pollution control as a proportion of GDP.

On the other hand, according to existing studies, external factors such as industrial structure (*IS*), population density (*PD*), technological innovation (*TI*), urban-rural structure (*UR*), and capital stock (*CS*) can also influence the spatial and temporal variation of GWF efficiency [3,12,30,38]. In this study, the industrial structure is represented by the share of tertiary industry in GDP; population density is tabulated as the ratio of total population to administrative district land area; technological innovation is represented by the number of patents granted; urban-rural structure is represented by the share of urban population in total population; the level of capital stock is represented by the fixed capital stock. The data on the above driving forces are obtained from *the China Statistical Yearbook*.

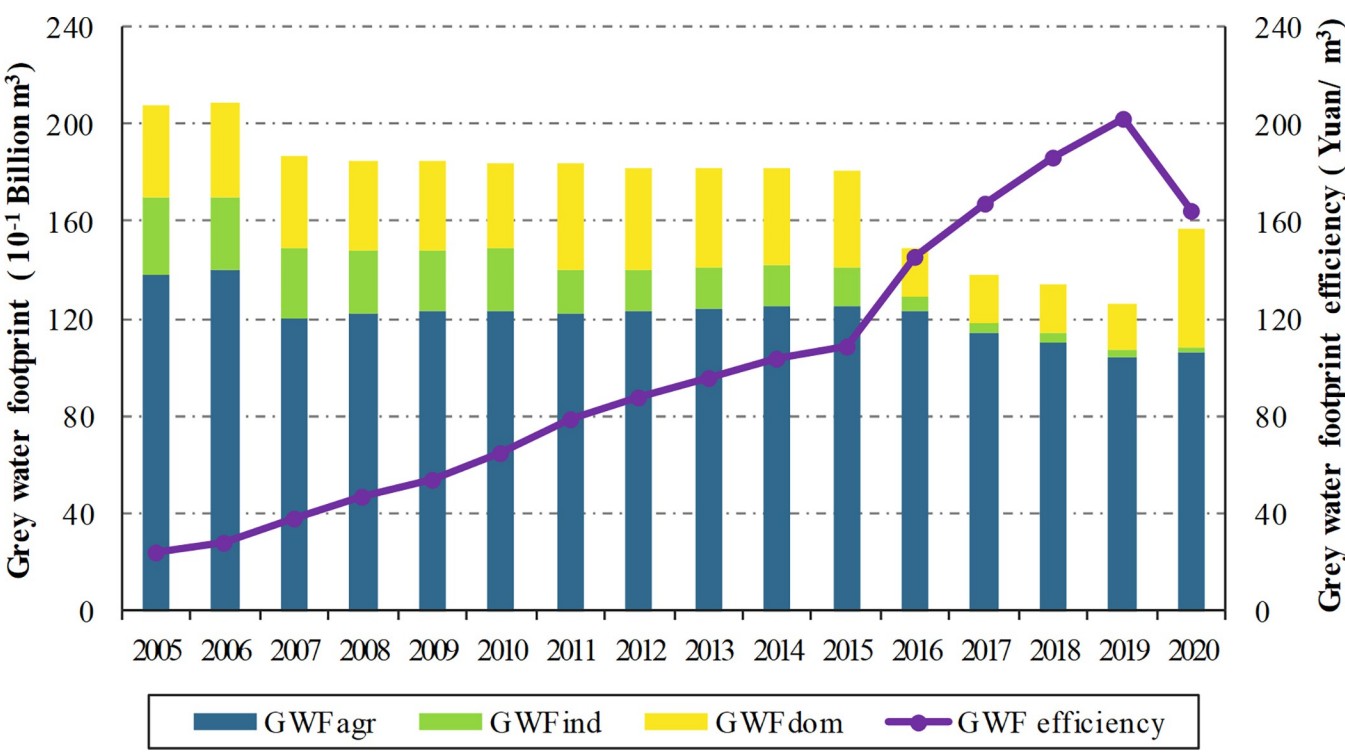

**Fig 2. Changes in the GWF and its efficiency in the Yellow River Basin from 2005 to 2020.**

## Results

### Measurement of GWF efficiency

The GWF of different sectors and the corresponding GWF efficiency in the Yellow River Basin can be calculated, as shown in Fig 2. As a whole, the GWF efficiency of the Yellow River Basin increases from 23.92 yuan/m$^3$ in 2005 to 164.87 yuan/m$^3$ in 2020. It is mainly due to the reduction of the GWF and the rapid growth of GDP. On the one hand, GDP rises from 570.26 billion yuan in 2005 to 2820.69 billion yuan in 2020. It indicates that the economy of the Yellow River Basin has achieved substantial development in recent years. On the other hand, the total GWF shows a fluctuating downward trend with a decline from 20.75 billion m$^3$ to 15.74 billion m$^3$ during the observation period. The year 2007 is a significant turning point, and since then, the GWF has switched from rising to gradually declining, with a rebound in 2020. This result is consistent with the studies by Zhang and Sun [38], and Lin et al. [44]. It is mainly associated with the implementation of "basin restrictions" since 2007, i.e., limiting the number of polluting enterprises and strengthening water quality supervision [45]. With the strengthening of environmental regulation and the upgrading of industrial standards, sewage discharge has been controlled. This result also indirectly reflects the effectiveness of China's environmental regulation policy. In addition, both the GWF and its efficiency change abruptly in 2020. Compared to 2019, the domestic GWF is significantly higher in 2020, leading to an increase in the total GWF. However, due to macro factors such as the New Crown epidemic, the growth in GDP of 2020 is slight, which results in a significant decrease in the GWF efficiency. It indicates that urban wastewater treatment in the basin needs to be strengthened. In response, in 2021, China's Development and Reform Commission and Ministry of Housing and Construction jointly issue the *Implementation Plan for the Urban Sewage and Waste Treatment in the Yellow*

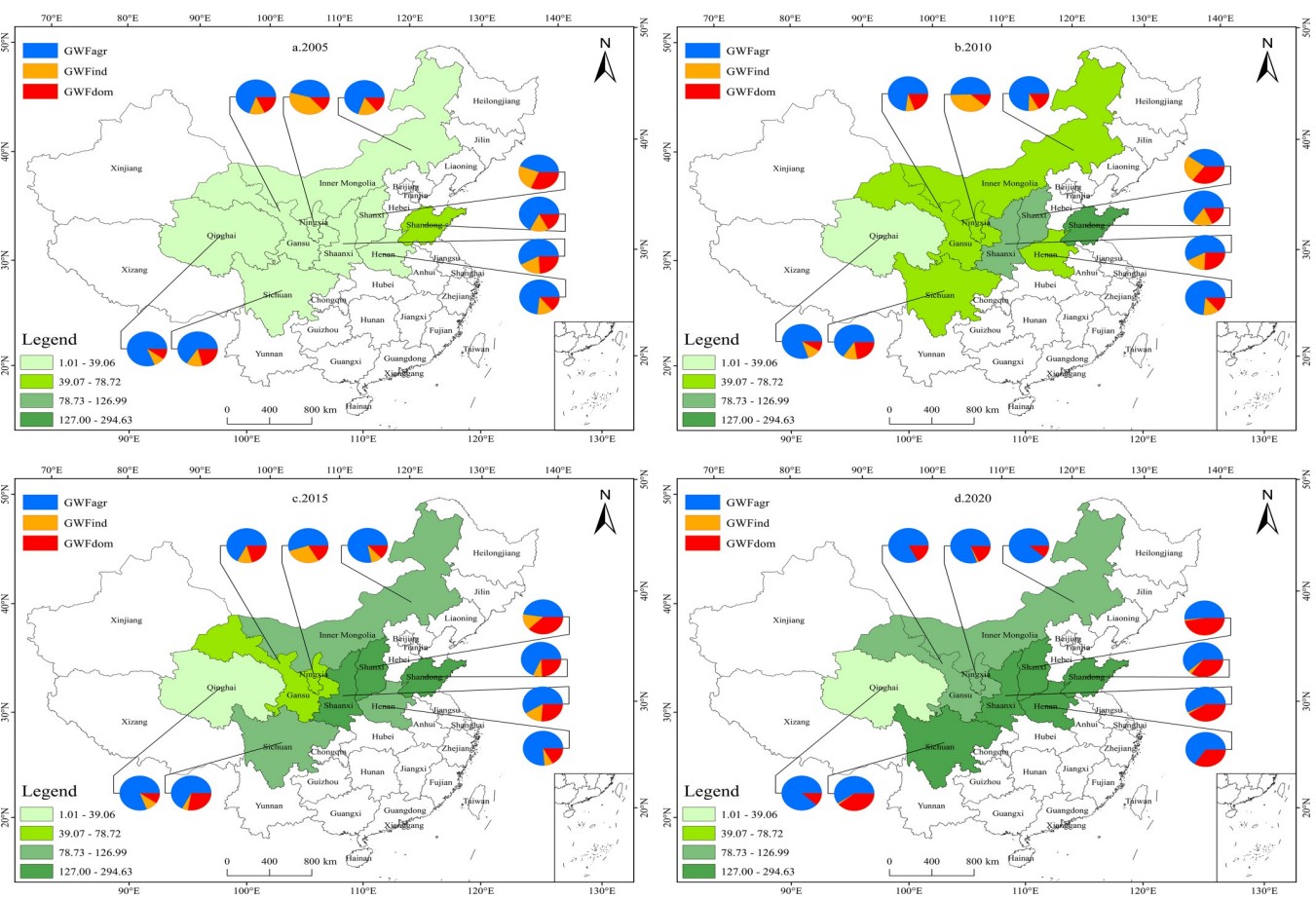

**Fig 3. GWF efficiency and the decomposition of GWF by province.**

*River Basin during the "14th Five-Year Plan" period* to improve the domestic sewage treatment rate.

To observe the changes in GWF efficiency of different provinces, Fig 3 reports the GWF efficiency and the proportion of the GWF of different sectors at four-time points: 2005, 2010, 2015 and 2020. As can be seen, the GWF efficiency varies considerably between provinces, showing a spatial distribution pattern of "low in the western and high in the eastern". The GWF efficiency of Shandong province in the downstream remains the highest, while Qinghai province in the upstream is the lowest. Qinghai is the source of the Yellow River and has a relatively good water environment, with a low GWF of 6.82 billion m$^3$, which is around 2/5 of the average GWF of the basin as a whole. However, as an economically underdeveloped region, Qinghai is still in the early stages of industrial development, and this situation has not been effectively improved over the years. The average GDP of Qinghai is only 186.14 billion yuan, much lower than the overall basin average GDP of 1733.16 billion yuan, which causes the GWF efficiency to be small. This result is also reflected in the study by Han et al. [10]. Differently, the other upstream provinces (Sichuan, Gansu and Ningxia) have received a faster increase in GWF efficiency, due to a slight decrease in grey water emissions and a rapid rise in GDP. In comparison, Shandong province in the downstream has secured an increase in GWF efficiency from 46.82 yuan/m$^3$ to 294.63 yuan/m$^3$ in parallel with stable socio-economic development. The reason is that Shandong, as an estuary province of the Yellow River, has a

comparative advantage and is better developed in terms of economy, industry and technology [3]. Especially, core cities in Shandong, such as Jinan, have made efforts to promote high-quality development, with the growth rate of major economic indicators remaining in the top echelon of the province and strategic emerging industries showing strong and resilient development trends.

In addition, the downstream province of Henan also experiences significant growth in GWF efficiency during the observation period, with an 8.24-fold increase. As a central agricultural province in the Yellow River Basin, Henan has accelerated its agricultural emission reduction efforts in recent years, and has been effective in controlling greywater emissions, decreasing from 45.01 billion $m^3$ to 28.38 billion $m^3$. Similarly, the GWF efficiency of Shanxi and Shaanxi in the midstream has risen rapidly, with increases of 230.85 yuan/$m^3$ and 206.64 yuan/$m^3$, respectively. However, Inner Mongolia, another midstream province, has seen relatively little improvement in the GWF efficiency. Despite its economic growth over the years, there has been no significant reduction in grey water emissions. Inner Mongolia has a sizeable east-west span, with a highly uneven distribution of resources and economic development. The western economic zone of Inner Mongolia, centered on Hohhot, Ordos and Baotou, is the main source of the province's economy, while the other regions have experienced very slow economic growth. Moreover, the industry based on energy sources such as coal has led to serious water pollution.

To further explore the heterogeneous role of different sectors, this study provides a detailed insight into the changes in agricultural, industrial and domestic GWF. Combining Figs 2 and 3, it is clear that the share of agricultural GWF is consistently the largest. From 2005 to 2020, the agricultural and industrial GWF fluctuate downwards, while the domestic GWF increases significantly. It indicates that agricultural grey water is the main source of grey water discharge in the Yellow River Basin. And over the years, grey water discharges from agriculture and industry have been controlled to some extent. In contrast to the actual situation, agricultural production in the Yellow River Basin is large in scale, with a wide distribution of rural areas, a concentrated population and a great number of livestock and poultry, which lead to severe non-point source pollution and a significant contribution to water pollutants [46,47]. It has caused the characteristic that agricultural GWF accounts for the largest share.

## Spatial and temporal variation of GWF efficiency

**Temporal variation.**   After the above analysis, it can be initially recognized that there is a certain variation in grey water footprint efficiency in the Yellow River Basin at the spatial and temporal dimensions. This study considers 2005, 2008, 2011, 2014, 2017 and 2020 as typical years and further explores the dynamics of temporal variation in GWF efficiency through the kernel density estimation method, as shown in Fig 4.

In terms of distribution position, the kernel density curve for GWF efficiency shifts to the right as a whole, showing a left-skewed distribution. It indicates that the GWF efficiency of the basin tends to be in an upward trend. In terms of the distribution pattern, the height of the main peak of the GWF efficiency curve decreases in a fluctuating manner, and the width becomes wider from 2005 to 2020. It means that the absolute difference in GWF efficiency is increasing and indirectly reflects the imbalanced development of the Yellow River Basin. For a long time, the Yellow River Basin has been characterized by a development pattern of "strong in the downstream and weak in the upstream". These findings reaffirm the significance of coordinated development in the Yellow River Basin. In terms of distribution extension, the curve of GWF efficiency has a clear left trailing feature. It demonstrates the existence of provinces with very poor GWF efficiency that should be controlled locally. In terms of polarization

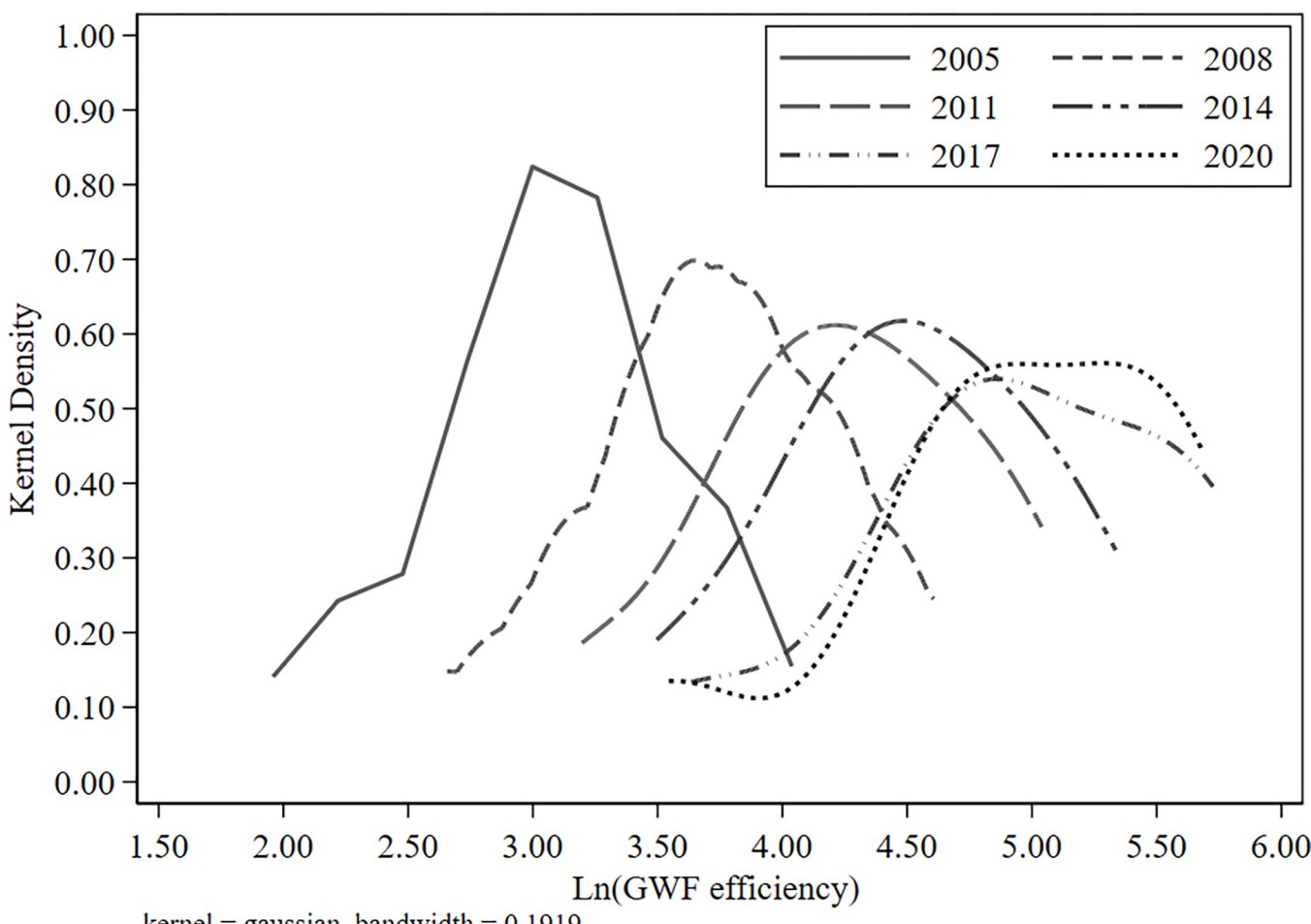

**Fig 4. Kernel density estimation of GWF efficiency.**

trends, the kernel density curve of GWF efficiency shows a multi-peaked pattern, with the main peak significantly higher than the side peaks. Although the pattern of the side peaks has gradually smoothed out in recent years, the overall trend has not changed. It suggests that the GWF efficiency in the Yellow River Basin is multi-polar. The variation between high and low provinces has decreased over the years, but the gradient effect is still significant. Therefore, it is necessary to implement differentiated water management in response to regional realities.

**Spatial variation.** To characterize the spatial variation of GWF efficiency in the Yellow River Basin, regional differences and sources of GWF efficiency are analyzed using the Dagum Gini coefficient and its decomposition method, and the results are presented in Table 1.

According to the results, the overall Gini coefficient of GWF efficiency in the Yellow River Basin shows a fluctuating upward trend during the observation period, rising from 0.25 in 2005 to 0.28 in 2020, with an increase of 12.83%. It indicates that the regional variation in GWF efficiency has become greater. Specifically, the average contribution of intra-regional variation, inter-regional variation and hypervariable density is 17.39%, 73.39% and 9.23%, respectively, with significant stratification. It can be seen that inter-regional variation is the main source of spatial variation in GWF efficiency in the Yellow River Basin. This result emphasizes the objective fact that the development of the Yellow River Basin is imbalanced once again. From the perspective of the contribution of variation, the contribution of inter-

**Table 1. Dagum Gini coefficient and decomposition.**

| year | Basin-wide Gini Coefficient | Intra-regional variation | | | Inter-regional variation | | | Contribution rate (%) | | |
|---|---|---|---|---|---|---|---|---|---|---|
| | | Upstream | Midstream | Downstream | Up-Mid | Up-Down | Mid-Down | Intra-regional | Inter-regional | Hypervariable density |
| 2005 | 0.25 | 0.17 | 0.10 | 0.15 | 0.18 | 0.41 | 0.32 | 17.17 | 76.38 | 6.46 |
| 2006 | 0.26 | 0.18 | 0.08 | 0.15 | 0.20 | 0.42 | 0.32 | 16.60 | 76.72 | 6.68 |
| 2007 | 0.27 | 0.17 | 0.10 | 0.17 | 0.20 | 0.45 | 0.35 | 16.73 | 77.45 | 5.82 |
| 2008 | 0.28 | 0.18 | 0.11 | 0.16 | 0.21 | 0.47 | 0.37 | 16.27 | 77.50 | 6.23 |
| 2009 | 0.28 | 0.19 | 0.08 | 0.14 | 0.21 | 0.46 | 0.38 | 14.83 | 76.98 | 8.19 |
| 2010 | 0.28 | 0.19 | 0.09 | 0.15 | 0.20 | 0.44 | 0.38 | 15.63 | 74.79 | 9.58 |
| 2011 | 0.28 | 0.19 | 0.10 | 0.15 | 0.19 | 0.44 | 0.41 | 15.88 | 73.26 | 10.87 |
| 2012 | 0.27 | 0.18 | 0.09 | 0.15 | 0.19 | 0.43 | 0.40 | 15.85 | 73.07 | 11.09 |
| 2013 | 0.27 | 0.19 | 0.09 | 0.16 | 0.19 | 0.43 | 0.39 | 16.00 | 73.34 | 10.66 |
| 2014 | 0.28 | 0.19 | 0.09 | 0.16 | 0.21 | 0.44 | 0.38 | 16.00 | 74.35 | 9.65 |
| 2015 | 0.28 | 0.19 | 0.11 | 0.17 | 0.20 | 0.44 | 0.37 | 17.15 | 73.64 | 9.21 |
| 2016 | 0.28 | 0.19 | 0.14 | 0.18 | 0.20 | 0.42 | 0.39 | 18.92 | 69.27 | 11.81 |
| 2017 | 0.30 | 0.14 | 0.20 | 0.22 | 0.20 | 0.43 | 0.41 | 20.63 | 67.63 | 11.73 |
| 2018 | 0.30 | 0.13 | 0.19 | 0.21 | 0.19 | 0.43 | 0.42 | 20.31 | 68.28 | 11.41 |
| 2019 | 0.30 | 0.09 | 0.19 | 0.24 | 0.17 | 0.44 | 0.42 | 20.39 | 68.91 | 10.69 |
| 2020 | 0.28 | 0.10 | 0.16 | 0.23 | 0.17 | 0.44 | 0.38 | 19.81 | 72.66 | 7.53 |

regional variation decreases from 76.38% to 72.66%. On the contrary, the contribution of intra-regional variation and hypervariable density rises, with an increase of 15.39% and 16.63%, respectively. It indicates a slight decrease in the inter-regional variation of GWF efficiency. However, it is still at a high level, far exceeding the sum of the other contributions.

This study further analyzes intra-regional and inter-regional variation separately. The intra-regional variation in both the upstream and midstream during the observation period is slight, with corresponding average Gini coefficients of 16.70%, 12.08% and 17.37%, respectively. Between 2005 and 2015, the intra-regional variation is consistently greatest in the upstream and smallest in the midstream. After 2015, the intra-regional Gini coefficient in the downstream rises significantly, leapfrogging the upstream. It indicates that the variation between Shandong Province and Henan Province in the downstream has been expanding in recent years. Inter-regional variation exhibits different characteristics. The average value of the inter-regional Gini coefficient for GWF efficiency in the Yellow River Basin is high and fluctuates very slightly. The inter-regional variation between the upstream and downstream is consistently the largest, while the variation between the upstream and midstream is the smallest. It is due to the large environmental and economic differences between the upper and lower reaches of the Yellow River Basin. Therefore, the integrated management of the Yellow River Basin should pay more attention to differentiation and develop relevant policies for different regions.

### Analysis of driving forces

**Internal driving forces.** In this study, q statistics and significance levels for each driving factor are calculated with the help of a factor detector (as seen in Table 2). The driving force of the internal factors of the GWF efficiency in the Yellow River Basin is then investigated. Fig 5 shows the driving force of the different internal factors at the spatial and temporal dimensions. It is seen in Fig 5(A) that economic development is the dominant factor in the evolution of GWF efficiency in the Yellow River Basin as a whole and the three major regions. It is mainly due to the fact that areas with poor economic development have smaller GDPs and are often

**Table 2. The driving force of internal factors on the GWF efficiency.**

| Region | Driving force | ED | WC | PL | LI | EI |
|---|---|---|---|---|---|---|
| Basin-wide | q statistic | 0.59 | 0.16 | 0.35 | 0.18 | 0.11 |
| | p value | 0.00 | 0.00 | 0.00 | 0.00 | 0.01 |
| Upstream | q statistic | 0.61 | 0.20 | 0.47 | 0.21 | 0.27 |
| | p value | 0.00 | 0.05 | 0.00 | 0.04 | 0.01 |
| Midstream | q statistic | 0.82 | 0.11 | 0.64 | 0.55 | 0.30 |
| | p value | 0.00 | 0.49 | 0.00 | 0.00 | 0.03 |
| Downstream | q statistic | 0.94 | 0.38 | 0.78 | 0.09 | 0.11 |
| | p value | 0.00 | 0.10 | 0.00 | 0.72 | 0.69 |

accompanied by problems such as sloppy production practices and high pollutant emissions, which inhibit GWF efficiency [10]. In addition, the pollution level is also a driving factor for the spatial and temporal variation of GWF efficiency that cannot be ignored. Specifically by region, for the basin as a whole and the upstream region, water consumption, labour input and environmental capital input contribute to improving GWF efficiency while the driving force is relatively small. In the midstream, labour input also has a significant role in the change of GWF efficiency except for economic development and pollution level. For the downstream, attention should also be paid to the water consumption factor. The downstream of the Yellow River suffers from severe water scarcity. Over the years, poor irrigation practices, rapid population growth and artificial diversions have led to reduced water availability and degradation of the water environment in the lower reaches, especially at the mouth of the delta [48,49]. Thus, improving the GWF efficiency in the downstream also requires rational water consumption management. According to the temporal evolution of the driving effects of different internal factors depicted in Fig 5(B), economic development has long been the main driver of the spatial and temporal variation in GWF efficiency, with the corresponding q statistics remaining high despite a significant decline after 2015. Except for the economic development factor, all other factors show relatively significant fluctuating changes.

The interaction detector is also used to explore the superimposed effects of the different internal driving factors, with the specific results shown in Fig 6. It can be found that the interaction of any two factors enhances the driving force of individual factors on changes of GWF efficiency, and that all combinations of interactions are either non-linearly strengthened or

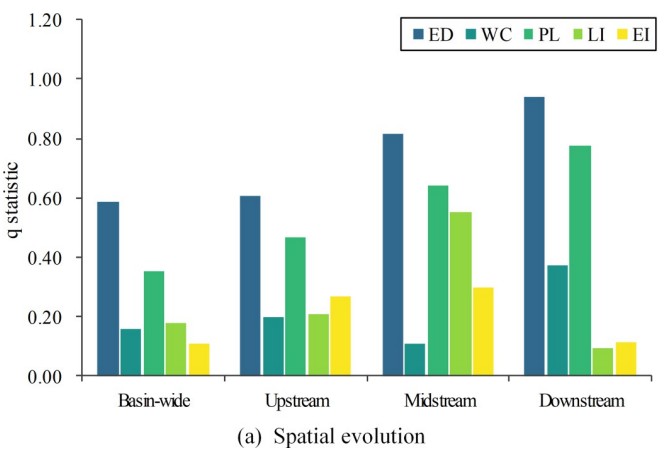

(a) Spatial evolution

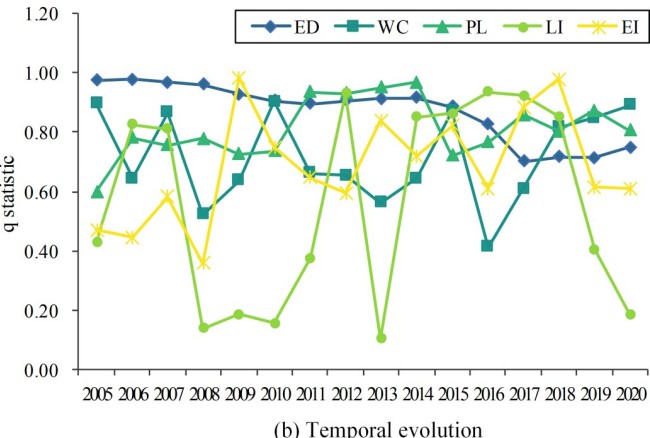

(b) Temporal evolution

**Fig 5. Spatial and temporal evolution of the driving force of internal factors.**

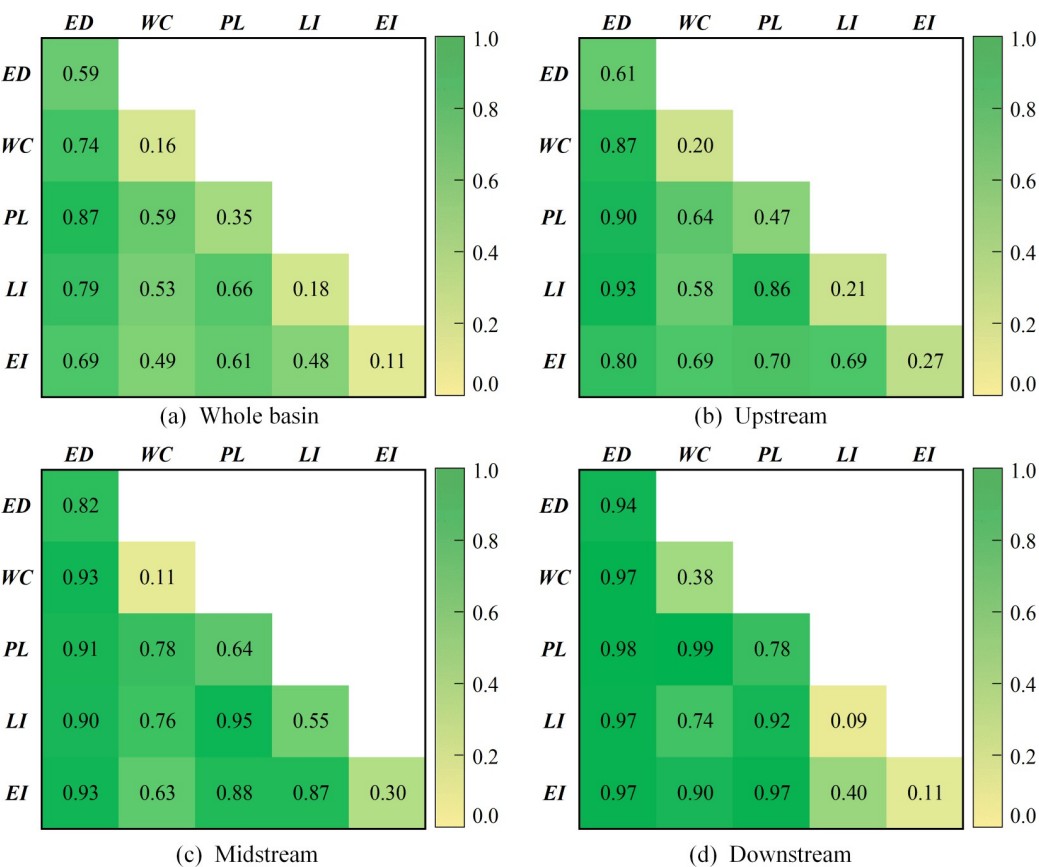

**Fig 6. Detection results of the interaction of internal factors.**

two-way strengthened. It suggests that the spatial and temporal variation in GWF efficiency receives the combined effect of multiple internal factors. By coordinating the relationship between the different factors, GWF efficiency can be effectively promoted. The interaction between economic development and other factors is always the highest for both the basin as a whole and the three sub-regions, especially in the downstream areas, where the driving force is over 90%. At the same time, the q statistics for the interaction between pollution level and other factors are all above 0.35. For the basin as a whole, the interaction between economic development and pollution level is the most significant, further suggesting that these two factors are critical to the GWF efficiency. Regarding the sub-region, the interaction between economic development and weakly driven labour input in the upstream has a significantly higher q statistic than the other combinations. It indicates that the improvement of GWF efficiency in the upstream should not only consider economic aspects, but also increase labour input and promote employment. For the midstream and downstream, when the water consumption factor with a weak driving force is combined with other factors, especially pollution level, the impact on GWF efficiency will be very significant. Therefore, controlling pollution emissions while using water resources wisely is also an effective initiative to promote GWF efficiency in the middle and lower reaches.

**External driving forces.** To explore the driving forces of the spatial and temporal variation in GWF efficiency, it is necessary to consider not only the direct effect of internal factors, but also the indirect driving force of external factors. As a result, this study further calculates the q statistics of the external factors and judges the interaction of different external factors.

**Table 3. The driving force of external factors on the GWF efficiency.**

| Region | Driving force | IS | PD | TI | UR | CS |
|---|---|---|---|---|---|---|
| Basin-wide | q statistic | 0.07 | 0.34 | 0.63 | 0.54 | 0.70 |
| | p value | 0.08 | 0.00 | 0.00 | 0.00 | 0.00 |
| Upstream | q statistic | 0.26 | 0.44 | 0.62 | 0.39 | 0.66 |
| | p value | 0.02 | 0.00 | 0.00 | 0.00 | 0.00 |
| Midstream | q statistic | 0.31 | 0.50 | 0.79 | 0.53 | 0.82 |
| | p value | 0.03 | 0.00 | 0.00 | 0.00 | 0.00 |
| Downstream | q statistic | 0.92 | 0.74 | 0.93 | 0.91 | 0.82 |
| | p value | 0.00 | 0.00 | 0.00 | 0.00 | 0.00 |

All the external driving forces reported in Table 3 have passed the 10% significance level test, confirming the high explanatory power of each external factor on the variation of GWF efficiency.

Similarly, Fig 7 illustrates the spatial and temporal evolution of the driving forces of external factors. It can be seen that capital stock is the main external driver of the variation in GWF efficiency, and its contribution remains at the highest position throughout the period 2005–2020. For all regions of the Yellow River Basin, it is important to use the power of capital to promote GWF efficiency. The driving force of technological innovation is also vital. The effect of technological innovation is not only reflected in the ability to improve the treatment rate of domestic and industrial wastewater, but also to improve fertilizer uptake and reduce fertilizer application, thereby mitigating agricultural pollution [27]. Furthermore, based on the temporal evolutionary characteristics of the driving forces of external factors, it is also clear that the driving force of population density on GWF efficiency increases rapidly after 2011. This is because areas with higher population accommodation capacity can effectively avoid the problem of excessive resource use or pollution emissions in zones of a certain size, reducing the environmental load on water bodies [50]. For the basin as a whole, the driving forces of the industrial structure and urban-rural structure are relatively weak; while in the downstream, the effect of these two factors is significant.

Similar to the internal factors, the interaction of any two external factors is significantly greater than the effect of individual factors (as shown in Fig 8). For the downstream region, the mean value of the q-statistic is as high as 0.93, indicating that the synergistic effect of

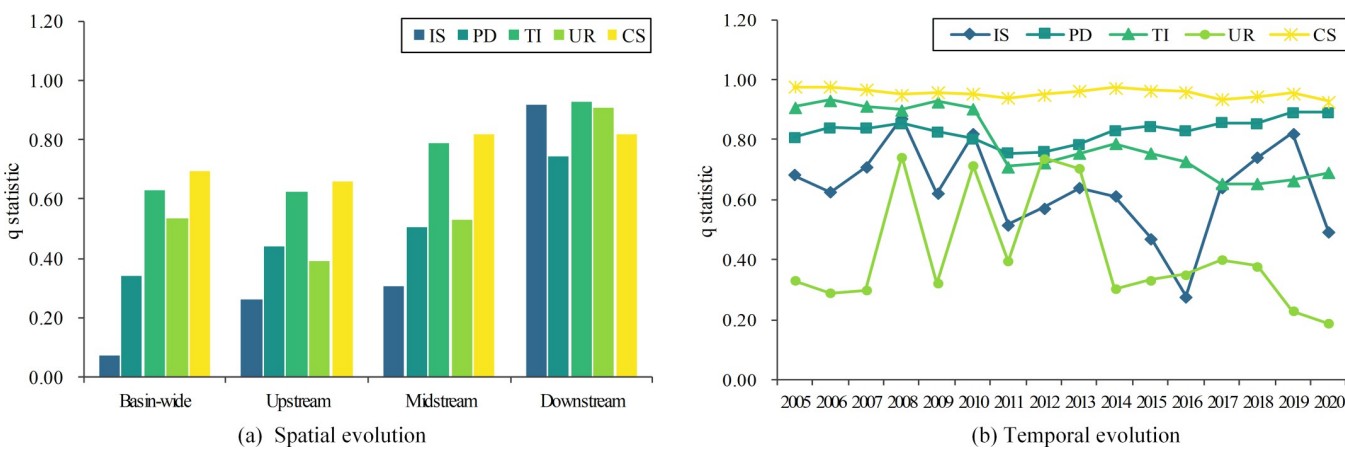

(a) Spatial evolution     (b) Temporal evolution

**Fig 7. Spatial and temporal evolution of the driving force of external factors.**

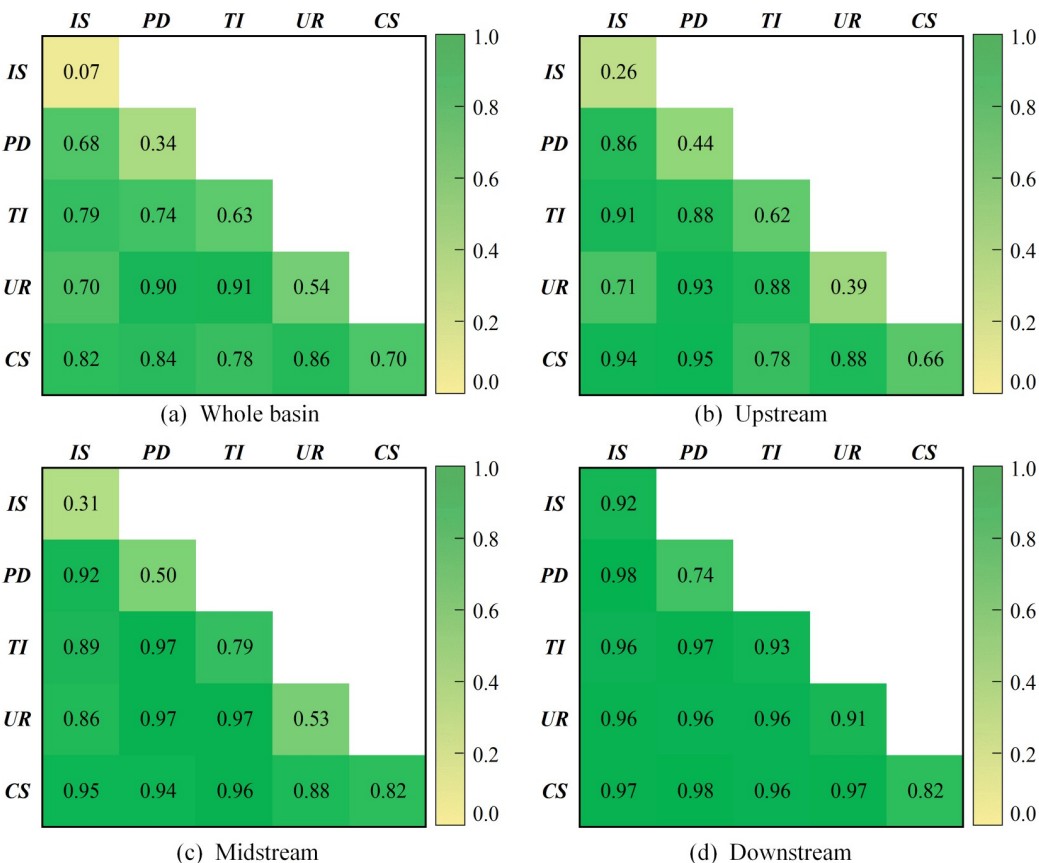

**Fig 8. Detection results of the interaction of external factors.**

external driving factors significantly enhances the explanation of the spatial and temporal variation in GWF efficiency in the Yellow River Basin. In terms of specific factors, the interaction of capital stock and technological innovation with other drivers is significant for the basin as a whole and for the three sub-regions. The dominant role of these two factors is again affirmed. In terms of sub-regions, the strongest combinations of q statistics corresponding to the upstream, midstream and downstream are capital stock and population density, technological innovation and population density, and industrial structure and population density, respectively. Therefore, each region should take locally appropriate measures to improve the GWF efficiency.

## Discussion

### Comparison with previous researches

This study measures the GWF and its efficiency for 9 provinces in the Yellow River Basin. The results have some commonalities with previous research which indicates the reasonableness of this study. For example, the GWF in this study decreases from 20.75 billion m$^3$ in 2005 to 15.74 billion m$^3$ in 2020. The year 2007 is a significant split point, and since then, the GWF has gradually changed from increasing to decreasing. Such results are consistent with the studies by Zhang and Sun [38], and Lin et al. [44]. They measure the GWF of 31 Chinese provinces and cities from 2000–2014 and 1998–2016, respectively, and similarly find that the GWF shifts from an increasing to a decreasing trend from the year 2007 onwards. This change mainly

stems from the implementation of environmental regulation policies at that time. In addition, Han et al. [10], Fu et al. [12], and Xu et al. [13] show that the GWF efficiency in China or the Yangtze River Basin is generally low in the west and high in the east, indirectly confirming the reliability of the results in this study.

However, the differences between the results of this study and other existing studies cannot be ignored. Specifically, Zhang et al. find that the domestic GWF makes the most considerable contribution to the total GWF, averaging about 69% [3], while this study concludes that agricultural GWF is the primary source of GWF in the Yellow River Basin. The reason for this discrepancy is that, on the one hand, the difference in study samples and intervals leads to some bias in the specific values measured. On the other hand, the Yellow River Basin is an important agricultural production area with a wide rural distribution, population concentration and a large number of livestock and poultry. It has led to serious non-point source pollution and water pollution [46,47].

Using the Geodetector method, this study also explores the internal and external driving forces of the spatial and temporal variation in GWF efficiency. It is found that economic development is the main internal driver of the variation in GWF efficiency. Han et al. [10], Zhang et al. [3], Fu et al. [12], Feng et al. [27], and Xu et al. [13] decompose the GWF or its efficiency using the Kaya equation and the LMDI model, and similarly find that the economic effect is an essential driving factor. In terms of external driving factors, capital stock and technological innovation are the primary drivers of GWF efficiency. Similar conclusions can be obtained in the studies by Zhang and Sun [38], Fu et al. [12], and Feng et al. [27]. Distinguishing from the existing literature, this study further investigates the interaction of different internal or external factors on the variation of GWF efficiency. It can be found that the interaction of any two internal or external factors is more significant than the effect of a single factor. Furthermore, the combinations of interactions with the highest q statistics vary for different regions. Therefore, it is necessary to take differentiated measures to improve grey water footprint efficiency for different regions.

## Policy implications

Based on the above analysis, this study suggests that synergistically improving the GWF efficiency of the Yellow River Basin requires not only a basin-wide perspective, but also disaggregated measures according to regional advantages.

Firstly, a region-wide coordinated development mechanism should be established to gradually reduce the absolute variation of GWF efficiency in the basin. Regions with high GWF efficiency should play a leading role, while low-efficiency regions should seek a new balance between environmental protection and economic development that is adapted to their demands. Secondly, the temporal and spatial differences in economic development and water pollution status among the three major regions of the basin should be fully considered, and initiatives should be differentiated to improve the GWF efficiency. For the upstream region, on the one hand, water environment regulation should be strengthened to enhance water containment capacity; on the other hand, for the main grain-producing areas such as the Hetao Plain and the Fenwei Plain, modern agriculture should be developed through technological innovation. Measures in the midstream region should be centered on pollution control and soil and water conservation. For example, controlling industrial pollution emissions, building silt dams and developing dry terracing. In the downstream, the focus can be on the protection of the delta area. Thirdly, the role of different internal and external driving forces should be brought into play. On the one hand, pay attention to the interaction between economic development and other factors. At the same time, it is necessary to accelerate the cultivation of strategic new

industries and promote the rational allocation of water resources and the upgrading of energy structures. On the other hand, the focus should be on safeguarding capital stock and resolving the conflict between people and land.

## Conclusion

This study introduces a comprehensive methodology for assessing and analyzing the GWF efficiency. Firstly, the GWF and its efficiency are calculated for the nine provinces of the Yellow River Basin from 2005 to 2020 from the agricultural, industrial and domestic perspectives. On this basis, the spatial and temporal variation in GWF efficiency is comprehensively investigated by the kernel density estimation and the Dagum Gini coefficient method, and the internal and external driving factors of the GWF efficiency are identified by the Geodetector method. The findings of the study are as follows.

1. Between 2005 and 2020, the GWF of the Yellow River Basin decreases from 20.75 billion $m^3$ to 15.74 billion $m^3$, and the agricultural GWF is the main contributor to GWF. In addition, the GWF efficiency increases from 23.92 yuan/$m^3$ to 164.87 yuan/$m^3$, showing a distribution pattern of "low in the western and high in the eastern". The GWF efficiency of Shandong province in the downstream remains at the highest level, while Qinghai province in the upstream is at the lowest.

2. The absolute temporal variation of GWF efficiency shows a rising trend. The kernel density curve is characterized by a significant left trailing and polarization, indicating the existence of areas with very low GWF efficiency and significant multi-polar differentiation in the Yellow River Basin. The spatial variation of GWF efficiency fluctuates upwards, accompanied by an increase in the overall Dagum Gini coefficient from 0.25 to 0.28. Inter-regional variation is the main source of spatial variation in GWF efficiency, with an average contribution of 73.39%. By sub-region, intra-regional variation is consistently higher in the upstream than in the midstream and downstream; the most serious spatial variation in GWF efficiency is found between the upstream and downstream.

3. In terms of internal driving forces, economic development is the primary driver of spatial and temporal variation in GWF efficiency in both the Yellow River Basin as a whole and the three regions. The interaction of any two internal factors is more significant than the effect of individual factors; in terms of external driving forces, the capital stock has the greatest impact on GWF efficiency. The corresponding interaction combinations with the highest q statistics for upstream, midstream and downstream are capital stock and population density, technological innovation and population density, and industrial structure and population density, respectively.

## Supporting information

**S1 File.**
(TXT)

**S1 Data.**
(XLSX)

## Acknowledgments

We would like to express our sincere thanks to the editors and anonymous reviewers for their constructive comments and suggestions.

## Author Contributions

**Data curation:** Yu Liu.

**Formal analysis:** Yun Li.

**Investigation:** Yu Liu.

**Methodology:** Lihua Yang.

**Resources:** Yu Liu.

**Software:** Lihua Yang.

**Visualization:** Yun Li.

**Writing – review & editing:** Tianbo Fu.

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
