## [Decision Letter · Decision Letter 0]

6 Dec 2022

PONE-D-22-25102Variation and internal-external driving forces of grey water footprint efficiency in China's Yellow River BasinPLOS ONE

Dear Dr. Liu,

Thank you for submitting your manuscript to PLOS ONE. After careful consideration, we feel that it has merit but does not fully meet PLOS ONE’s publication criteria as it currently stands. Therefore, we invite you to submit a revised version of the manuscript that addresses the points raised during the review process.

We look forward to receiving your revised manuscript.

Kind regards,

Xianzhong Cao

Academic Editor

PLOS ONE

Journal Requirements:

2. Please note that PLOS ONE has specific guidelines on code sharing for submissions in which author-generated code underpins the findings in the manuscript. In these cases, all author-generated code must be made available without restrictions upon publication of the work. Please review our guidelines at https://journals.plos.org/plosone/s/materials-and-software-sharing#loc-sharing-code and ensure that your code is shared in a way that follows best practice and facilitates reproducibility and reuse. New software must comply with the Open Source Definition.

“This work is supported by “the Graduate Research and Innovation Projects of Jiangsu Province” (grant number KYCX21_0441) and “the Fundamental Research Funds for the Central Universities” (grant number B220203024).”

“This work is supported by “the Graduate Research and Innovation Projects of Jiangsu Province” (grant number KYCX21_0441) and “the Fundamental Research Funds for the Central Universities” (grant number B220203024).”

“This work is supported by “the Graduate Research and Innovation Projects of Jiangsu Province” (grant number: KYCX21_0441; funder: Lihua Yang) and “the Fundamental Research Funds for the Central Universities” (grant number: B220203024; funder: Lihua Yang).Lihua Yang is the first author of this manuscript.”

7. We note that  Figure 3 in your submission contain map images which may be copyrighted. All PLOS content is published under the Creative Commons Attribution License (CC BY 4.0), which means that the manuscript, images, and Supporting Information files will be freely available online, and any third party is permitted to access, download, copy, distribute, and use these materials in any way, even commercially, with proper attribution. For these reasons, we cannot publish previously copyrighted maps or satellite images created using proprietary data, such as Google software (Google Maps, Street View, and Earth). For more information, see our copyright guidelines: http://journals.plos.org/plosone/s/licenses-and-copyright.

      1. You may seek permission from the original copyright holder of Figure 3 to publish the content specifically under the CC BY 4.0 license. 

Reviewers' comments:

Reviewer's Responses to Questions

**Comments to the Author**

1. Is the manuscript technically sound, and do the data support the conclusions?

Reviewer #1: Yes

2. Has the statistical analysis been performed appropriately and rigorously? 

Reviewer #1: Yes

3. Have the authors made all data underlying the findings in their manuscript fully available?

Reviewer #1: Yes

4. Is the manuscript presented in an intelligible fashion and written in standard English?

Reviewer #1: Yes

5. Review Comments to the Author

Reviewer #1: This manuscript investigated the spatial and temporal variation of the grey water footprint (GWF) efficiency based on the measurement of the GWF efficiency, the kernel density estimation and the Dagum Gini coefficient method. The paper is written in sound English, but with some grammar errors. Detailed comments are as follows:

1. “For demonstrating this assessment approach, China's Yellow River Basin is chosen as an example.”, spatial and temporal scales should be clarified for the empirical study in this sentence.

2. “Inter-regional variation is the 41 primary source of spatial variation, with an average contribution of 73.39%.”, please clarify inter-regional variation of what?

3. A map should be added to provide a detailed description of the study area.

4. As to the numbers in the text and tables, I suggest the authors keep only two decimal places in a number.

5. In methods and data, the use of operator signs in formulas should be uniform.

6. Natural and social conditions differ significantly within provinces, such as Inner Mongolia. Is the provincial-scale analysis too general? If possible, analysis at a more precise scale could be more convincing.

7. According to Fig. 1, both TGWF and GWF efficiency were mutated in 2020. An analysis of this phenomenon can be added to the text.

8. The graphic style of Fig. 3 does not show well the three-dimensional distribution of information. Try adjusting the angle or changing the layout to make the information clear.

9. Some grammar problems can be checked.

6. PLOS authors have the option to publish the peer review history of their article (what does this mean?). If published, this will include your full peer review and any attached files.

Reviewer #1: No

---

## [Author Response · Author response to Decision Letter 0]

17 Dec 2022

The response to the reviewers' comments has been uploaded to the system.

---

## [Decision Letter · Decision Letter 1]

6 Mar 2023

Variation and internal-external driving forces of grey water footprint efficiency in China's Yellow River Basin

PONE-D-22-25102R1

Dear Dr. Liu,

We’re pleased to inform you that your manuscript has been judged scientifically suitable for publication and will be formally accepted for publication once it meets all outstanding technical requirements.

Kind regards,

Xianzhong Cao

Academic Editor

PLOS ONE

Additional Editor Comments (optional):

Reviewers' comments:

Reviewer's Responses to Questions

**Comments to the Author**

1. If the authors have adequately addressed your comments raised in a previous round of review and you feel that this manuscript is now acceptable for publication, you may indicate that here to bypass the “Comments to the Author” section, enter your conflict of interest statement in the “Confidential to Editor” section, and submit your "Accept" recommendation.

Reviewer #2: (No Response)

2. Is the manuscript technically sound, and do the data support the conclusions?

Reviewer #2: Yes

3. Has the statistical analysis been performed appropriately and rigorously? 

Reviewer #2: Yes

4. Have the authors made all data underlying the findings in their manuscript fully available?

Reviewer #2: Yes

5. Is the manuscript presented in an intelligible fashion and written in standard English?

Reviewer #2: Yes

6. Review Comments to the Author

Reviewer #2: The authors comprehensively revised the manuscript according to the comments of reviewers. I have only small suggestions for the manuscript as follows:

1. Please indicate the units of variables in the method and data parth.

2. In the results section, it can be seen from Figure 2 that in 2016, GWF decreased significantly and the efficiency of GWF increased significantly. Please explain the reason briefly.

3. Line 425, 76.37% should be 76.38%.

4. In the internal driving forces section, Please describe the p value in Table 2.

7. PLOS authors have the option to publish the peer review history of their article (what does this mean?). If published, this will include your full peer review and any attached files.

Reviewer #2: No

---

## [Editor Report · Acceptance letter]

14 Mar 2023

PONE-D-22-25102R1 

Variation and internal-external driving forces of grey water footprint efficiency in China's Yellow River Basin 

Dear Dr. Liu:

I'm pleased to inform you that your manuscript has been deemed suitable for publication in PLOS ONE. Congratulations! Your manuscript is now with our production department. 

Kind regards, 

on behalf of

Professor Xianzhong Cao 

Academic Editor

PLOS ONE